# Nodule Characteristics, Clinical Risk Factors, and Radiologist Experience as Predictors of Positive Baseline LDCT Screening Results

**DOI:** 10.3390/healthcare13212734

**Published:** 2025-10-29

**Authors:** Jelena Djekic Malbasa, Tomi Kovacevic, Marija Vukoja, Daliborka Bursać, Darijo Bokan, Vladimir Stojšić, Bojan Zaric

**Affiliations:** 1Faculty of Medicine, University of Novi Sad, 21000 Novi Sad, Serbia; tomi.kovacevic@mf.uns.ac.rs (T.K.); marija.vukoja@mf.uns.ac.rs (M.V.); daliborka.bursac@mf.uns.ac.rs (D.B.); darijo.bokan@mf.uns.ac.rs (D.B.); vladimir.stojsic@mf.uns.ac.rs (V.S.); 2Institute for Pulmonary Diseases of Vojvodina, 22000 Sremska Kamenica, Serbia

**Keywords:** lung cancer, low-dose computed tomography (LDCT), lung-RADS, pulmonary nodules, screening, predictors

## Abstract

**Background/Objectives:** Early detection of lung cancer through low-dose computed tomography (LDCT) significantly improves patient outcomes. Identifying predictors of positive baseline LDCT findings can optimize screening programs and cost-effectiveness, particularly in regional settings. **Methods:** We conducted a retrospective analysis of baseline LDCT screenings performed in 2024 at three sites in Vojvodina, Serbia. Eligible participants were adults aged 50–74 years with a significant smoking history. Lung nodules were evaluated using the Lung-RADS system. Sociodemographic, clinical, and nodule-related variables, as well as radiologist experience (>10 vs. <10 years), were assessed. Multivariable logistic regression identified independent predictors of positive LDCT outcomes (Lung-RADS 3, 4A, 4B, 4X). **Results:** Overall, 17.6% (613/3479) of participants had positive baseline LDCT findings. Multivariable analysis showed that nodule type (semi-solid: OR = 4.01, 95% CI: 1.46–11.01; solid: OR = 8.86, 95% CI: 4.53–17.32), nodule morphology (smooth: OR = 0.42, 95% CI: 0.22–0.81; perifissural: OR = 0.16, 95% CI: 0.08–0.33; endobronchial: OR = 47.99, 95% CI: 12.35–186.58), nodule diameter (OR = 1.83 per mm, 95% CI: 1.71–1.96), presence of COPD (OR = 1.95, 95% CI: 1.23–3.08), age (OR = 1.02 per year, 95% CI: 1.00–1.04), and radiologist <10 years working experience (OR = 1.47, 95% CI: 1.23–3.08) were independent predictors of positivity. **Conclusions:** Baseline LDCT positivity is influenced by nodule characteristics, patient sociodemographic and clinical factors, and radiologist experience. These findings can inform targeted screening strategies in high-risk populations.

## 1. Introduction

Lung cancer remains the leading cause of cancer-related mortality worldwide, with almost 2.5 million new cases and over 1.8 million deaths reported in 2022 [1]. Despite significant advances in thoracic oncology, the overall 5-year survival rate remains low—approximately 20%—ranging from 65% for early-stage disease to only 5% for advanced-stage cancer. This poor prognosis is mainly because about 70% of new lung cancer cases are diagnosed at an advanced stage, when treatment options are limited and less effective [2,3].

Randomized controlled trials (RCTs), including the National Lung Screening Trial (NLST), the Dutch–Belgian NELSON trial, and the Multicentric Italian Lung Detection (MILD) trial, have demonstrated that low-dose computed tomography (LDCT) screening can significantly reduce lung cancer mortality by 20%, 25%, and 39%, respectively. This benefit is primarily attributed to a stage shift, with a greater proportion of cases—up to 70%—detected at an early, more treatable stage [4,5,6]. Meta-analyses of these RCTs further confirm a 17% relative reduction in lung cancer-specific mortality and a 4% reduction in overall mortality associated with LDCT screening [7,8]. These findings have prompted the implementation of LDCT screening programs in many regions worldwide; however, differences in organizational structure, inclusion criteria, and resource allocation persist [9].

In Europe, only a few countries have implemented national LDCT lung cancer screening programs, including Croatia (2020), the Czech Republic (2022), and Poland (2023) [9,10]. Most other countries are still conducting regional pilots or feasibility studies [9]. To accelerate implementation and ensure equity across Europe’s heterogeneous healthcare systems, the European Union (EU) launched the SOLACE project (Strengthening the Screening of Lung Cancer in Europe) in 2023, supporting member states in developing structured and evidence-based lung cancer screening strategies [9,11].

Globally, lung cancer screening programs vary in inclusion criteria regarding age, tobacco exposure, and other risk factors, including environmental or occupational exposures, chronic respiratory conditions, and family history of lung cancer. Tobacco use, however, remains the predominant selection criterion across most programs [9,11,12].

Interpretation of LDCT findings is guided by the Lung Imaging Reporting and Data System (Lung-RADS), which standardizes reporting and management recommendations based on nodule size, type, morphology, and growth rate [13]. The Lung-RADS system has been validated in both research and clinical settings for its utility in improving screening performance and reducing false-positive rates [13,14,15]. While Lung-RADS primarily emphasizes radiological features, multiple studies have shown that patient-related factors—such as age, smoking history, and comorbidities, particularly chronic obstructive pulmonary disease (COPD)—also influence the likelihood of a positive screening result [9,14,15,16,17].

Baseline screening rounds are particularly informative, as they typically yield higher positivity rates than subsequent rounds, reflecting the underlying prevalence of pulmonary nodules in the screened population [14]. Recent analyses have highlighted the impact of radiologist-related factors. Prior studies demonstrated that radiologist experience was independently associated with lower false-positive rates, suggesting that human factors contribute substantially to variability in screening outcomes [18,19]. Similarly, interobserver agreement improves with the use of standardized tools and training, but differences across centers persist [20,21]. This highlights the importance of considering both patient demographics and screening site-level variables when assessing predictors of positive LDCT findings.

The LDCT pilot screening program in Vojvodina, Serbia, has operated since 2020 as the country’s first organized initiative. While some preliminary findings, such as emphysema prevalence, have been published [22], the prevalence and risk factors for positive baseline LDCT findings have not been systematically evaluated. This study provides the first comprehensive analysis of predictors of positive baseline LDCT outcomes in this region. By identifying independent predictors (Lung-RADS 3, 4A, 4B, 4X), it aims to address a key knowledge gap and inform more efficient, cost-effective screening in resource-limited settings.

## 2. Materials and Methods

### 2.1. Study Design and Setting

This retrospective study analyzed baseline LDCT screening data collected during 2024 from three screening sites in Vojvodina, Serbia: one tertiary healthcare/academic hospital (n = 938) and two general hospitals (n = 1486 and n = 1055). All sites followed a standardized LDCT screening protocol and uniform reporting procedures to ensure data comparability across centers.

### 2.2. Study Population

Eligible participants were individuals aged 50–74 years, asymptomatic and without clinical signs of lung cancer. Inclusion criteria encompassed current smokers with a smoking history of ≥30 pack-years or ≥20 pack-years with additional risk factors (COPD, recurrent pneumonia in adulthood (≥18 years of age), prior malignancy at another site, family history of lung cancer, or occupational exposure to carcinogens), and former smokers who had quit within the past 10 years with ≥20 pack-years.

### 2.3. Exclusion Criteria

Temporary exclusions applied to individuals with recent lower respiratory tract infections (LDCT postponed for 4–6 weeks or until recovery) or those who had undergone chest CT within the previous 12 months. Permanent exclusions included participants diagnosed with lung cancer within the past five years or those with advanced comorbidities limiting expected survival.

### 2.4. Lung Nodule Assessment

Lung nodule evaluation was performed according to the Lung-RADS (Lung Imaging Reporting and Data System) classification, which standardizes LDCT reporting and management recommendations [13,14,23]. The Lung-RADS version 1.1 (released in 2019) has been applied consistently since the initiation of the pilot screening program in Vojvodina and remains in use to ensure methodological consistency across all study phases.

Negative screening results (categories 1–2) indicate either the absence of nodules or the presence of benign-appearing nodules, such as small (<6 mm) solid or ground-glass nodules, calcified granulomas, or perifissural nodules consistent with intrapulmonary lymph nodes. Positive screening results (categories 3, 4A, 4B, and 4X) include nodules that are probably or highly suspicious for malignancy and require short-term follow-up LDCT or additional diagnostic evaluation.

For each participant, the dominant nodule was characterized by its type (solid, part-solid, non-solid/ground-glass opacity [GGO], or calcified) and morphology (endobronchial, smooth-contour, perifissural, or spiculated). Clinical and demographic variables, including age, sex, smoking exposure (pack-years), presence of COPD, and radiologist experience, were also analyzed with respect to LDCT outcomes.

### 2.5. Data Collection and Quality Assurance

Before LDCT screening, all participants completed a standardized questionnaire on sociodemographic data, comorbidities, family history of lung cancer, smoking history, and occupational exposure. The questionnaire was administered and completed by trained healthcare professionals involved in the pilot program, rather than by patients themselves. Information on COPD (ICD-10 code J44) was obtained from primary care electronic records. In line with national clinical practice, this diagnosis is routinely confirmed by spirometry performed by general practitioners, although historical spirometric data were not available for verification in this study. Radiologists at each site received standardized training on LDCT screening protocols, structured reporting, and Lung-RADS–based management. All personal data were de-identified prior to analysis, and confidentiality was maintained in accordance with the Declaration of Helsinki and international data protection standards.

### 2.6. Statistical Analysis

Descriptive statistics were calculated for all variables. Continuous variables are expressed as mean ± standard deviation (SD) or median (interquartile range, IQR), as appropriate, while categorical variables are presented as counts and percentages.

Comparisons between positive and negative screening results were made using the chi-square test for categorical variables and the Mann–Whitney U test for continuous variables. Multivariable logistic regression identified independent predictors of positive LDCT outcomes (Lung-RADS 3–4X). Variables included in the model—nodule type, nodule morphology, age, sex, smoking history (pack-years), presence of COPD, and radiologist experience—were selected based on clinical relevance and univariate analysis. Odds ratios (ORs) with 95% confidence intervals (CIs) were calculated. Potential multicollinearity among covariates was assessed using pairwise correlations, Tolerance, and Variance Inflation Factors (VIFs). No variable exceeded commonly accepted thresholds (Tolerance < 0.2 or VIF > 5), indicating no concerning collinearity. The potential non-linearity of continuous predictors was further assessed by including centered quadratic terms as a sensitivity analysis to confirm model robustness. All statistical analyses were performed using SPSS version 26, with two-sided *p* < 0.05 considered statistically significant.

### 2.7. Ethical Approval

This study was approved by the Ethics Committee of the Institute for Pulmonary Diseases of Vojvodina, Sremska Kamenica, Serbia (No. 110-V/1). Written informed consent was obtained from all participants.

## 3. Results

### 3.1. Baseline Characteristics

Among the 3479 participants analyzed, 613 (17.6%) exhibited positive LDCT findings (Lung-RADS 3–4X) (Table 1). Participants with positive results were generally older, more frequently male, and had greater cumulative smoking exposure (all *p* < 0.05). This group also demonstrated slightly lower BMI and a higher prevalence of COPD. No significant differences were identified for smoking status, history of pneumonia, family history, education, or marital status. The highest proportion of positive results was observed among retirees.

### 3.2. Nodule Characteristics

Nodule type and morphology demonstrated strong associations with LDCT positivity (Table 2). Semi-solid and solid nodules predominated among positive findings, whereas calcified nodules were primarily associated with negative results (*p* < 0.001). Spiculated and endobronchial nodules exhibited the highest probability of positivity compared to smooth and perifissural types. The mean nodule diameter and volume were significantly greater in the positive group (both *p* < 0.001).

### 3.3. Screening Site and Radiologist Experience

LDCT positivity rates were comparable between academic and community hospital sites. Radiologists with less than 10 years of experience, however, reported positive findings more frequently than those with 10 or more years of experience (*p* = 0.013, Table 3).

### 3.4. Multivariable Logistic Regression

The multivariable logistic regression model identified several independent predictors of positive LDCT results (Table 4).

Older age, presence of COPD, nodule type (semi-solid and solid compared to calcified), nodule morphology (endobronchial, smooth, and perifissural compared to spiculated), and larger nodule diameter were each significantly associated with increased odds of positivity.

Radiologist experience remained an independent predictor, suggesting that reader expertise influences reporting patterns.

Endobronchial nodules, while rare, demonstrated a particularly strong association with positivity due to the high rate of positive classifications within this subgroup.

Other demographic and exposure-related variables did not reach statistical significance in the adjusted model.

## 4. Discussion

In this regional, real-world cohort of 3479 baseline LDCT screening examinations from three screening sites in Vojvodina, Serbia, 17.6% were classified as positive (Lung-RADS 3, 4A, 4B, or 4X). Independent predictors of a positive screen included nodule features (type, morphology, and diameter), patient characteristics (older age and COPD), and radiologist experience.

Reported baseline positivity rates vary widely across RCTs and real-world programs due to differences in inclusion criteria, image acquisition protocols, and nodule management algorithms [7,9,12,14,15]. The NLST, which applied a simple size threshold (≥4 mm), reported baseline positivity around 27% with high false-positive rates [4,18], whereas re-analysis of NLST data using Lung-RADS reduced positivity to approximately 12–13% [24]. In contrast, the NELSON trial, which used volumetric thresholds and interval rescanning, reported a markedly lower baseline positivity rate (2.6%) [5]. Similarly, the MILD trial reported 14% baseline positivity using volume-based cut-offs for non-calcified nodules (>250 mm^3^) with specific follow-up protocols [6,25]. Real-world U.S. programs using Lung-RADS have reported positivity rates ranging from 10.7% to 24.6%, depending on population risk and screening setting, with higher rates in community hospitals than in tertiary centers [26,27,28,29]. This heterogeneity in design and management protocols largely explains the wide variation in baseline positivity across studies.

The baseline positivity rate of 17.6% (17.9% in community and 16.7% in academic hospitals) in our study, slightly higher than in most real-world programs, can be explained by several interrelated factors. First, Lung-RADS v1.1 (2019) [23] was used to maintain consistency with the original pilot screening protocol; this version classifies airway (endobronchial) nodules as category 4A, potentially increasing positive rates relative to the updated v2022 system [13]. Second, our cohort compared to major RCTs, included a higher proportion of women (56.8% vs. 0–45% in RCTs) and active smokers (86.4% vs. 48–77% in RCT) [4,5,6]. Although mean tobacco exposure was similar to that in the NELSON trial [5], the higher prevalence of current smokers likely contributed to the increased positivity rate, given the association between smoking and both malignant and inflammatory CT findings. Third, emphysema, which was highly prevalent in this population, is independently associated with nodule detection and higher Lung-RADS categories [22]. Finally, two of the three screening sites were general hospitals newly implementing screening in 2024; during this early implementation phase, radiologists likely applied a more cautious interpretive approach to avoid missing potentially malignant lesions. Taken together, these factors explain the relatively high baseline positivity rate observed in this pilot phase. As radiologist experience increases and screening protocols mature—incorporating updated Lung-RADS versions and refined training—positivity rates are expected to gradually stabilize.

Consistent with prior evidence [23,25,30,31], imaging features—especially the presence of a solid component and a larger diameter—were the strongest predictors of a positive scree in our adjusted model. A slight non-linear trend was observed, suggesting a modest plateau at larger diameters; such patterns are biologically plausible, as the incremental predictive yield of size tends to diminish beyond certain thresholds. This minor deviation did not materially affect the interpretation of diameter as a robust and independent predictor. Solid nodules (OR 8.86 vs. calcified nodules) and semisolid/part-solid nodules (OR 4.01) had markedly higher odds of being scored positive, while non-solid (GGO) nodules showed a non-significant association. These results align with prior reports that part-solid and solid components confer substantially higher malignancy risk than purely ground-glass or calcified lesions, and that size remains a key continuous predictor for malignancy probability [30,31].

Morphology beyond attenuation added independent information: spiculation was associated with higher odds of a positive result, whereas smooth contours and perifissural morphology were associated with much lower odds. The lower risk associated with perifissural nodules is well described; many of these represent benign intrapulmonary lymph nodes or scars and are frequently downgraded in structured management algorithms [31].

Endobronchial nodules represented a small subgroup (1.7%) but had an extremely high positivity rate (95%). Although rare, their inclusion substantially influenced the odds ratio. Prior studies report a low prevalence (~0.5%) with the majority of endobronchial opacities resolving on repeat imaging, indicating predominantly benign outcomes [32]. Nevertheless, NELSON data indicate that up to 22% of missed cancers later present as central or endobronchial lesions [33], underscoring the need for careful airway evaluation. Lung-RADS v2022 provides refined guidance for these findings, incorporating lesion persistence, multiplicity, and internal attenuation [13]. Given the limited number of cases and potential differences in local reading practices, these results should be interpreted with caution; the wide confidence intervals likely reflect statistical instability rather than a true causal effect.

Beyond imaging, clinical variables added independent predictive value. In our multivariable analysis, older age and a diagnosis of COPD were associated with higher odds of a positive LDCT screen, in line with NLST findings [4,18]. The link with COPD is biologically and epidemiologically plausible, as airflow obstruction and emphysema increase both true cancer risk and the frequency of inflammatory or indeterminate nodules. Previous studies have shown that COPD is common and elevates lung cancer risk independently of smoking exposure [15,16,17,34]. No concerning collinearity was observed between COPD and smoking exposure (pack-years), confirming their independent contributions to screening positivity. In NLST, other factors such as female sex, white race, heavier smoking history, marital status, hard rock mining, and farm work were also independently associated with a positive screen [18].

Radiologist experience remained an independent predictor in multivariable analysis. Radiologists with <10 years of experience were more likely to classify findings as positive, consistent with prior studies showing higher false-positive rates among less experienced readers [18,19,34]. Across the three participating sites, a total of 16 radiologists were involved in baseline screening, reflecting the diversity of experience typical for early program implementation. Although the <10 vs. ≥10 years dichotomization is a simplified proxy for expertise, it provides a pragmatic and comparable framework for evaluating performance in a real-world setting. Before participation, all radiologists underwent standardized training at the tertiary center, and all positive findings were subsequently reviewed in multidisciplinary meetings. Computer-aided detection (CAD/AI) tools were routinely implemented across all sites. The observed association likely reflects both the learning curve inherent to early program phases and expected inter-reader variability at baseline. Future research should incorporate radiologist-level variables—such as structured training metrics, reading workload, and double reading—to better understand performance variability. These findings collectively provide a framework for optimizing screening implementation and serve as a foundation for the forthcoming national program.

While our results are consistent with major randomized trials, this study adds important real-world evidence from a middle-income healthcare system during the early phase of LDCT screening implementation. The observed associations between patient-, nodule-, and radiologist-related factors reflect both screened cohort characteristics and operational realities that differ from controlled trial conditions. These findings offer practical insights for adapting screening protocols, inclusion criteria, and training frameworks to regional contexts and resource-limited settings. Program-level monitoring and site-level quality assurance remain essential to balance sensitivity and specificity. Standardized training, structured reporting (e.g., Lung-RADS), and integration with smoking-cessation and multidisciplinary care pathways are key to ensuring consistency and cost-effectiveness [18,19,20,35]. As emphasized by Wait et al. (2022), a health-systems approach with adaptable performance indicators is needed to translate trial-based evidence into sustainable and equitable European screening programs [35].

Baseline rounds yield the highest positivity rates and should guide planning for referral capacity and diagnostic workload [9,15,35]. Combining imaging features (attenuation, part-solid component, spiculation, size) with clinical factors (age, COPD, smoking status) may further improve risk prediction, while validated volumetry-based models can help reduce false positives [9,13].

This study has several limitations. It is confined to a single baseline year and three regional sites, which may limit generalizability. COPD diagnoses were derived from primary-care records (ICD-10 J44) without spirometry verification, potentially underestimating prevalence. Variability among radiologists and across sites is inherent to early implementation, and detailed radiologist-level data (case volume, training hours, double reading) were not captured. The small number of endobronchial nodules yielded wide confidence intervals and should be interpreted with caution. Finally, the lack of longitudinal follow-up precludes assessment of false-positive resolution, interval cancers, and program-level performance metrics; future multi-round analyses will be essential to evaluate long-term clinical and economic outcomes.

## 5. Conclusions

In this regional, real-world baseline cohort, nodule imaging features—particularly solid or part-solid composition and larger size—were the strongest predictors of positive LDCT results. Clinical factors (age and COPD) and radiologist experience provided additional predictive value. The relatively high positivity rate observed reflects both population characteristics and the early phase of program implementation. These findings underscore the need for risk-adapted selection and management strategies, ongoing radiologist training, and the integration of AI-assisted tools to improve specificity, standardization, and cost-effectiveness in lung cancer screening programs.

## Figures and Tables

**Table 1 healthcare-13-02734-t001:** Sociodemographic and clinical characteristics in relation to LDCT screening outcomes.

Variable	Total (n = 3479)	Lung-RADS Positive (n = 613)	Lung-RADS Negative (n = 2866)	*p*-Value
Age, mean (SD)	62.21 (6.55)	64.01 (6.24)	61.83 (6.57)	<0.001 *
Sex, n (%)				0.039
Male	1504 (43.2)	288 (19.1)	1216 (80.9)	
Female	1975 (56.8)	325 (16.5)	1650 (83.5)	
BMI, mean (SD)	26.72 (4.77)	26.18 (4.76)	26.83 (4.76)	0.005 *
Pack-years, mean (SD)	38.68 (18.21)	41.23 (19.97)	38.13 (17.77)	<0.001 *
Active smoker, yes (%)	3005 (86.4)	532 (17.7)	2473 (82.3)	0.744
COPD, yes (%)	210 (6.1)	63 (30.0)	147 (70.0)	<0.001
COPD, no (%)	3269 (93.9)	550 (16.8)	2719 (83.2)	
Pneumonia, yes (%)	163 (4.7)	24 (14.7)	139 (85.3)	0.320
Family history of lung cancer, yes (%)	1212 (34.8)	375 (17.2)	1808 (82.8)	0.562
Employment, n (%)				<0.001
Employed	518 (14.9)	189 (14.0)	1160 (86.0)	
Unemployed	1349 (38.8)	81 (15.6)	437 (84.4)	
Retired	1612 (46.3)	343 (21.3)	1269 (78.7)	

Notes: Values are presented as mean (SD) or n (%). *p*-values were derived from χ^2^ tests (categorical variables) or Mann–Whitney U tests (continuous variables), as appropriate. * *p* < 0.05 was considered statistically significant. Abbreviations: BMI—body mass index; COPD—chronic obstructive pulmonary disease; LDCT—low-dose computed tomography; Lung-RADS—Lung Imaging Reporting and Data System.

**Table 2 healthcare-13-02734-t002:** Nodule characteristics in relation to LDCT screening outcomes.

Variable	Total (n = 3479)	Lung-RADS Positive (n = 613)	Lung-RADS Negative (n = 2866)	*p*-Value
Nodule type, n (%)				<0.001
Calcified	268 (7.7)	15 (5.6)	253 (94.4)	
Non-solid GGO	183 (5.3)	33 (18.0)	150 (82.0)	
Semi-solid	78 (2.2)	27 (34.6)	51 (65.4)	
Solid	1582 (45.5)	505 (31.9)	1077 (68.1)	
Morphology, n (%)				<0.001
Endobronchial	59 (1.7)	56 (9.7)	3 (5.1)	
Smooth	1469 (42.2)	354 (24.1)	1115 (75.9)	
Perifissural	435 (12.5)	69 (15.9)	366 (84.1)	
Spiculated	125 (3.6)	97 (77.6)	28 (22.4)	
Nodule diameter, mean (SD), mm	3.53 (4.714)	8.45 (7.93)	2.48 (2.70)	<0.001 *
Nodule volume, mean (SD), mm^3^	127.28 (626.35)	556.44 (1401.98)	35.50 (92.45)	<0.001 *

Notes: Values are presented as mean (SD) or n (%). *p*-values were derived from χ^2^ tests (categorical variables) or Mann–Whitney U tests (continuous variables) *, as appropriate. *p* < 0.05 was considered statistically significant. Abbreviations: GGO—ground-glass opacity.

**Table 3 healthcare-13-02734-t003:** Screening site and radiologist experience in relation to LDCT screening outcomes.

Variable	Total (n = 3479)	Lung-RADS Positive (n = 613)	Lung-RADS Negative (n = 2866)	*p*-Value
Screening site, n (%)				0.407
Academic	938 (26.9)	157 (16.7)	781 (83.3)	
Community hospital	2541 (73.1)	456 (17.9)	2085 (82.1)	
Radiologist experience, n (%)				0.013 *
<10 years	734 (21.1)	152 (20.7)	582 (79.3)	
>10 years	2745 (78.9)	461 (16.8)	2284 (83.2)	

Note: *p*-values derived from χ^2^ tests (categorical variables) or Mann–Whitney U tests (continuous variables) *, as appropriate. *p* < 0.05 was considered statistically significant.

**Table 4 healthcare-13-02734-t004:** Multivariable logistic regression analysis of predictors of positive LDCT screening results.

Variable	Category	OR (95% CI)	*p*-Value	Reference
Sex	Male	0.93 (0.71–1.22)	0.608	Female
BMI	Per unit	0.98 (0.95–1.00)	0.088	-
Age	Per year	1.02 (1.00–1.04)	0.039	-
Pack-years	Per unit	1.00 (0.99–1.01)	0.271	-
Nodule type	Calcified		<0.001	Reference
	Non-solid GGO	1.95 (0.81–4.68)	0.135	Calcified
	Part-solid	4.01 (1.46–11.01)	0.007	Calcified
	Solid	8.86 (4.53–17.32)	<0.001	Calcified
Morphology	Spiculated		<0.001	Reference
	Endobronchial	47.99 (12.35–186.58)	<0.001	Spiculated
	Smooth	0.42 (0.22–0.81)	0.009	Spiculated
	Perifissural	0.16 (0.08–0.33)	<0.001	Spiculated
Diameter	Per mm	1.83 (1.71–1.96)	<0.001	-
COPD	Present	1.95 (1.23–3.08)	0.004	Absent
Radiologist experience	<10 years	1.47 (1.09–1.98)	0.012	≥10 years

Notes: Continuous predictors are expressed per unit increase (diameter per mm; age per year; BMI per unit; pack-years per unit). Reference categories: sex (Female), nodule type (Calcified), morphology (Spiculated), COPD (Absent), radiologist experience (≥10 years). Diagnostics: No relevant multicollinearity was observed (all VIF < 3.5; Tolerance > 0.3). Adding a centered quadratic term for nodule diameter slightly improved global fit (Δ − 2LL = 13.1; *p* < 0.001), indicating a mild deviation from linearity. The effect of diameter remained stable, and the variable was retained as continuous in the final model.

## Data Availability

The datasets presented in this article are not readily available because they are part of an ongoing project and subject to privacy restrictions. Requests to access the datasets should be directed to the Secretariat for Health Care of Vojvodina.

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
