# Peer review of "Nodule Characteristics, Clinical Risk Factors, and Radiologist Experience as Predictors of Positive Baseline LDCT Screening Results"

_healthcare, 2025, doi:10.3390/healthcare13212734_

Round 1
Reviewer 1 Report
Comments and Suggestions for Authors
The manuscript entitled “Nodule Characteristics, Clinical Risk Factors, and Radiologist Experience as Predictors of Positive Baseline LDCT Screening Results” is an attempt to analyse the trends of lung cancer/ low-dose computed tomography in the selected geographic area. However, the quality of the manuscript can be improved by addressing the following queries:
- Paper title: Abbreviation from the title, in this case LDCT, may be expanded for better clarity to the wider audience/readers.
- In the Introduction section, lines 33-34, statistics of lung cancer are too old, need to be updated with recent data for relevance.
- In the introduction section, line 54, expand the abbreviation of term SOLACE for clarity.
- In page 2, at the end of the introduction section, lines 87-93, authors have mentioned the implementation of the program. If some publication of this previous work may be cited and what is the research gap to be achieved through this present study should be discussed for better understanding of the study’s context and novelty.
- In the Materials and Methods section, please define the terms/codes first: Lung-RADS 3, 4A, 4B, 4X and thereafter it can be used throughout the manuscript for reader comprehension.
- In the Results section 3.1, what was the status of the brain health for the COPD cases? If this information is available, it can also be discussed for additional insights.
- In Table 3, Typos in the caption may be corrected for accuracy.
- In the discussion section, lines 337-340, authors have mentioned the limitations of the present studies. However, authors should also mention the key findings of the present studies with a scope of future studies for better contextualization and impact.
Reviewer 2 Report
Comments and Suggestions for Authors
The paper analyzes baseline LDCT lung cancer screening across three sites in Vojvodina, Serbia (2024), using Lung-RADS to classify nodules and multivariable logistic regression to identify predictors of a positive baseline screen (defined as Lung-RADS 3–4X).
My major points:
- The paper reports 613/3,476 positives in the abstract section but 3,479 total in Results/Table 1. Please reconcile and propagate the corrected denominator everywhere.
- Methods cite Lung-RADS materials but don’t state the exact version used for 2024 reads (v1.1 vs v2022)! Given changes in airway findings management in v2022, please specify the version, decision rules, and how airway nodules were handled (initial and follow-up).
- The diameter effect size (OR 1.83 per mm) is extremely steep across the observed range and may reflect non-linearity or collinearity with nodule type/morphology.
- Please report missingness for each covariate (e.g., COPD; you later acknowledge incomplete spirometry) and your imputation or complete-case strategy. If COPD was clinician-reported without spirometry, state that clearly and quantify.
- Language: mostly clear, but there are typos/grammar issues (“Radiologies expirience,” “expirience,” punctuation in tables). A language/technical edit will help.
Reviewer 3 Report
Comments and Suggestions for Authors
The paper investigates factors associated with positive low-dose CT (LDCT) screening outcomes in a regional cohort in Serbia. The authors retrospectively analyzed 3479 participants screened in 2024. They report that nodule characteristics (type, morphology, and diameter), patient factors (age, COPD), and radiologist experience were independent predictors of positive LDCT findings. The study highlights the importance of both clinical and human factors in interpreting LDCT and aims to support more efficient and targeted lung cancer screening strategies.
The paper is interesting and the dataset is valuable. However, I believe the manuscript requires major revision before it can be considered for publication. Below are my detailed comments:
1. The study confirms findings already reported in NLST, NELSON, and other trials. The manuscript should better emphasize what is truly new in this regional Serbian cohort, beyond confirming existing predictors.
2. The analysis is restricted to one baseline year. This limits understanding of follow-up outcomes, false positives, or cancer confirmation. Authors should acknowledge more clearly how this weakens the conclusions.
3. The inclusion and exclusion criteria are described, but more detail on missing data handling (for example, COPD diagnosis without full spirometry) is required.
4. The logistic regression includes many variables, but potential multicollinearity (e.g., between smoking history and COPD) is not addressed.
5. Odds ratios for rare subgroups (endobronchial nodules) are unstable and should be interpreted with caution.
6. While experience was found to be significant, the analysis is crude (cut-off at 10 years). More nuanced measures (training, workload, second reading, use of CAD/AI) could improve interpretation. At least, the limitations of this simplification should be discussed.
7. The discussion compares positivity rates with international trials, but sometimes superficially. Stronger explanation of why the Serbian cohort shows higher positivity is needed.
8. Tables are informative but dense. The text sometimes repeats figures in tables rather than focusing on interpretation. Authors should simplify to improve readability.
Reviewer 4 Report
Comments and Suggestions for Authors
This is a retrospective study of regional, real-world cohort on the identifying independent predictors of positive baseline LDCT finding. Authors demonstrated that baseline LDCT positivity is influenced by nodule characteristics, patient sociodemographic and clinical factors, and radiologist experience. This study may provide some useful information on the targeted screening strategies in high-risk populations. I have some comments.
<Comments>
- In line 41, Please describe the full term of the abbreviation, “LDCT”.
- In line 54, Please describe the full term of the abbreviation, “EU”.
- In line 71, Please describe the full term of the abbreviation, “COPD”.
- In line 107, “chronic obstructive pulmonary disease (COPD)”
Please switch to COPD.
- In line 108, “adulthood”
Please describe in detail for the definition of adulthood. (ex, age > 18 year)
- In line 125, Please describe the full term of the abbreviation, “GGO”.
- How many radiologists attend to this study? And how many radiologists was experienced more than 10 years? Please describe this.
Round 2
Reviewer 3 Report
Comments and Suggestions for Authors
The authors addressed all my concerns. The paper can be accepted for publication.